# The first nationwide study on facing and solving ethical dilemmas among healthcare professionals in Slovenia

Štefan Grosek[1,2,3], Rok Kučan[4], Jon Grošelj[5], Miha Oražem[6], Urh Grošelj[7], Vanja Erčulj[8,9], Jaro Lajovic[8], Ana Borovečki[10], Blaž Ivanc[11] *

1 Division of Surgery, Department of Paediatric Surgery and Intensive Therapy, University Medical Centre Ljubljana, Ljubljana, Slovenia, 2 Chair of Paediatrics, Faculty of Medicine, University of Ljubljana, Ljubljana, Slovenia, 3 Neonatology Section, Division of Obstetrics and Gynaecology, Department of Perinatology, University Medical Centre, Ljubljana, Slovenia, 4 Faculty of Medicine, University of Ljubljana, Ljubljana, Slovenia, 5 Theological Faculty, University of Ljubljana, Ljubljana, Slovenia, 6 Department of Radiation Oncology, Ljubljana Institute of Oncology, Ljubljana, Slovenia, 7 Department of Paediatric Endocrinology, Diabetes and Metabolic Diseases, University Children's Hospital, UMC Ljubljana, Ljubljana, Slovenia, 8 Rho Sigma Research & Statistics, Ljubljana, Slovenia, 9 Faculty of Criminal Justice and Security, University of Maribor, Slovenia, 10 Andrija Štampar School of Public Health, School of Medicine, University of Zagreb, Zagreb, Croatia, 11 Faculty of Health Sciences, University of Ljubljana, Ljubljana, Slovenia

* blaz.ivanc@zf.uni-lj.si

**Data Availability Statement:** All relevant data are within the paper and its Supporting Information files.

## Abstract

### Background

Healthcare professionals (HCPs), patients and families are often faced with ethical dilemmas. The role of healthcare ethics committees (HECs) is to offer support in these situations.

### Aim

The primary objective was to study how often HCPs encounter ethical dilemmas. The secondary objective was to identify the main types of ethical dilemmas encountered and how HCPs solve them.

### Subjects and methods

We conducted a cross-sectional, survey-based study among HCPs in 14 Slovenian hospitals. A questionnaire was designed and validated by HCPs who were selected by proportional stratified sampling. Data collection took place between April 2015 and April 2016.

### Results

The final sample size was n = 485 (385 or 79.4%, female). The response rates for HCPs working in secondary and tertiary level institutions were 45% and 51%, respectively. Three hundred and forty (70.4%) of 485 HCPs (very) frequently encountered ethical dilemmas. Frequent ethical dilemmas were waiting periods for diagnostics or treatment, suboptimal working conditions due to poor interpersonal relations on the ward, preserving patients' dignity, and relations between HCPs and patients. Physicians and nurses working in secondary level institutions, compared to their colleagues working in tertiary level institutions, more

**Funding:** The study was funded as part of the research program Terciar—Project development of the Committee for Medical Ethics University Medical Centre Ljubljana and Clinical Ethics (No: 20140215) of the University Medical Centre Ljubljana in the years 2014–2016. The funders had no role in study design, data collection and analysis, decision to publish, or preparation of the manuscript.

**Competing interests:** The authors have declared that no competing interests exist.

frequently encountered ethical dilemmas with respect to preserving patients' dignity, protecting patients' information, and relations between HCPs and patients. In terms of solutions, all HCPs most frequently discussed ethical dilemmas with co-workers (colleagues), and with the head of the department. According to HCPs, the most important role of HECs is staff education, followed by improving communication, and reviewing difficult ethical cases.

## Conclusions

Waiting periods for diagnostics and treatment and suboptimal working conditions due to poor interpersonal relations are considered to be among the most important ethical issues by HCPs in Slovenian hospitals. The most important role of HECs is staff education, improving communication, and reviewing difficult ethical cases.

## Introduction

Technological innovations in medicine along with social movements in the United States and later in Europe in the 1950s and 1960s have brought about new and different ethical challenges in healthcare institutions that invariably involve tension between a healthcare professional's (HCP's) personal values with the institutional, legal, ethical or personal values of others and society [1–4]. If not properly addressed, this may ultimately result in personal moral distress [5, 6] and/or political and legal decision-making conflicts in the most severe cases [7–10]. In the era of rationing, constraints on financial and human resources, and changes in hospital management, these new constraints put even more pressure on an HCP's values with respect to what is right or wrong [11, 12]. Apart from renowned public cases on difficult ethical and mostly legal aspects in medicine [7–10], surveys among HCPs on confrontation with ethical dilemmas and ways of resolving them showed that medical science and clinical medicine professionals have to work together with ethicists, legal institutions, and public domains.

In December 2019, a Medline search query with the key word "ethical dilemmas" returned around 4,500 hits.

Hospital or institutional ethics committees were first introduced in the United States and later in Europe and other countries. Hospital ethics committees (HECs) have had a tremendous impact on recognising, improving, and resolving diverse ethical dilemmas. In the United States, HECs have existed since the early 1970s [13]. Only later were HECs introduced in countries throughout Europe as well as globally [14–23]. In addition to HECs, other means of resolving ethical dilemmas developed, mostly in northern Europe, where HCPs receive professional training to become facilitators of moral case deliberations among co-workers with whom they work every day [24, 25].

In Slovenia, resolving ethical dilemmas is still in the hands of the HECs and no official ethics training is currently available for HECs. Even in some difficult cases, other countries, i.e., with well-established HECs, recognised that physicians should be more active when they are faced with situations in which people may be affected due to their ignorance [26].

In Slovenia, the first HEC was established at the Faculty of Medicine, University of Ljubljana, in the second half of the1960s after the Declaration of Helsinki on human research came into effect [27, 29]. It was among the first in Europe, and its primary role was to evaluate the ethical adequacy of medical research projects by postgraduate students [28]. In the Faculty of Medicine, ethics has been taught since 1948 [29]. However, HECs were only introduced in

the late 1990s when ethics committees were already well established and functioning elsewhere in Europe. Only a small number of committees were, in fact, working on a regular basis, and not all hospitals had their own ethics committee. According to current information, HECs are now established and working regularly in all Slovenian hospitals.

In everyday clinical work, it is of utmost importance that the recommendations of HECs are implemented consistently and in line with other hospital services for maintaining good clinical practice [30–32].

HECs must be evaluated to establish whether their work has an impact, and which of the envisioned functions (e.g., individual case consultations, education of HCPs, and policy formation) do HECs actually perform in everyday clinical work [17, 33–36].

A motivation for this study was to evaluate how HCPs face ethical dilemmas and how they resolve them. This is the first national survey of HCPs (physicians, nurses, and other profiles of HCPs) working in 14 hospitals (secondary and tertiary level institutions). The primary objective was to study how often HCPs encounter ethical dilemmas. The secondary objective was to identify the main types of ethical dilemmas they face and how they solve them. We hypothesized that HCPs are frequently faced with ethical dilemmas. We also assumed that the type of ethical dilemmas encountered, the approaches used for ethical education, the awareness of the existence of HECs, and the extent to which the presence of HECs is considered relevant to the resolution of ethical dilemmas, are likely to differ among HCPs from different hospitals. A special aspect of this study was to establish whether there are differences between secondary and tertiary healthcare institutions in terms of what types of ethical dilemmas are most prevalent and how HCPs solve these dilemmas.

## Methods

### Overall design of the study

We conducted a cross-sectional, physical, survey-based study of HCPs (physicians, nurses, and other HCPs) in all ten secondary level general hospitals, two special hospitals for treatment of lung diseases (the secondary level Topolšica Hospital, and the tertiary level Golnik University Hospital for Lung Diseases), and in both tertiary level University Medical Centres in Ljubljana and Maribor, that is, in a total of 14 Slovenian hospitals. We prepared a written questionnaire (see S1 Appendix) designed for HCPs who were selected by the proportional stratified sampling method.

### Study context and participants

Based on results from a prior pilot study [37], we expected 60% of HCPs to have (very) often faced ethical dilemmas. To detect the effect with an accuracy of 5% at the significance level $\alpha = 0.05$ and with 80% power, 770 HCPs were required in the study. The expected non-response rate of 30% increased the sample size by an additional 230 HCPs.

The inclusion criterion was that participants were HCPs (physicians, nurses) and other -HCPs, (laboratory technicians and engineers, radiological engineers, clinical psychologists, nurse assistants, biochemical technicians and engineers, pharmacists, social workers, physiotherapists, respiratory therapists, speech therapists, hygiene technicians, and psychologists).

Due to the different professional and academic workloads and obligations of HCPs and other HCPs between secondary and tertiary level institutions, we expected that different ethical dilemmas would be perceived within each group and subgroup of HCPs and other HCPs.

Proportional stratified sampling was used to select HCPs for the study. An anonymized list of HCPs with their unique IDs was sent from each of the hospitals included in the study. We computed the proportion of HCPs to be included in the sample for each hospital. The

employees included in the sample were selected based on simple random sampling (the number of seed selection units in the sample was 02031979). Simple random sampling was performed with the R software package [38] via the call to the function "sample" and with random seed set to the date of the received list of HCPs.

## Data collection

All 11 Slovenian public secondary hospitals and three tertiary level hospitals were included in the study consecutively, one by one. The first hospital included in the study was the UMC Ljubljana. The data was collected from the UMC Ljubljana between April and July 2015. The data from the other hospitals were collected in the autumn and winter of 2015 and the spring of 2016.

The complete list of employees was obtained from the Human Resources Department of each hospital. According to the decision No. 090–59 / 2009 of the Information Commissioner dated July 13, 2009, a public employee is not entitled to privacy with regard to their names. Thus, the personal information for each employee could be acquired from the Human Resources Department of each hospital, after which the employee could decide whether to participate in the survey or not. The list was arranged in alphabetical order according to employees' last names. We informed the head and the head nurse of the clinical department of the hospitals by telephone and later by e-mails about the objectives of this research. The questionnaires were delivered to the administrative office of all clinical departments and wards of each hospital personally or by internal mail. Departmental secretaries were asked to distribute the questionnaires to the selected HCPs. In small hospitals, the internal mail was sent directly to HCPs. The responses to the questionnaires were then collected and put into the designated envelopes. These were collected after two weeks and put into a larger box. In this way, we ensured the complete anonymity of survey participants.

In some hospitals, only the personal registration numbers and professional profiles of HCPs were disclosed. Therefore, after we chose eligible HCPs according to their personal registration, the Human Resources Department in each of those hospitals distributed the questionnaire to HCPs by using their personal registration number.

## Questionnaire form

The questionnaire was study-specific (see S1 Appendix). It consisted of 20 questions, 8 of which were aimed at obtaining demographic information about the respondents.

The questions were divided into three parts. In the first part of the questionnaire, we asked for demographic data (age, gender, information about their profession, workplace, and work experience). In the second part, we aimed to determine how often HCPs are faced with ethical dilemmas and to estimate how they encounter ethical dilemmas in the domains listed in the questionnaire in their professional work. We were interested mainly in how they solved the recognised ethical dilemmas and what were the most important areas of responsibility of the HECs. The third part consisted of questions about the HEC in their institution. We were interested in finding out what percentage of HCPs were aware of the existence of the HEC. In the second and third parts of the questionnaire, the respondents answered four out of the 12 questions using a five-point Likert scale with frequency labels. In four questions, the respondents were asked to indicate "yes", "do not know" or "no" as their answer. The remaining four were multiple-choice questions with an option of an additional written response. The questionnaire was anonymous.

The questionnaire took about 10 minutes to complete. The questionnaire was accompanied by a text that explained the background and purpose of the study.

## Validation and testing of the questionnaire

The seven-step approach to questionnaire development, as recommended by the AMEE guidelines, was followed [39]. We first reviewed the literature, including in the scope of our research knowledge of the key ethical dilemmas found in the main tertiary hospital, the University Medical Centre Ljubljana, where four of the authors are members of the Hospital Ethics Committee and daily encounter various ethical issues raised by healthcare professionals. Afterwards, we synthesised the literature and interviews and developed the questionnaire. In the next step, we included a pre-test of the questionnaire on 35 HCPs at the University Medical Centre Ljubljana (UMC Ljubljana) to optimise the measurement instrument. Based on the pre-test results, we adjusted the sample size required for measuring the primary endpoint with a predetermined precision. We also removed those questions that were not answered at all during pretesting and showed a lack of measurement sensitivity. Please, see the whole validation and testing the questionnaire in the Supplementary Information file (S1 File).

## Statistics

The mean and standard deviations were calculated for continuous variables and frequencies and percentages for categorical variables. Answers on the Likert-type 5-point scale items dichotomized into two categories. The answers "frequently" and "very frequently" formed the first answer category, and the remainder, the second answer category. The proportion of HCPs (very) frequently encountering an ethical dilemma was calculated and the 95% CI was obtained using the bootstrap method. The relationship between an HCP's profile and the binary outcome variables were analysed using a mixed-effect logistic regression model to account for the clustering effect of individuals within hospitals. A mixed-effect linear regression model was used to analyse the relationship between a worker's profile and numerical outcome variables. The significance level threshold was set to $\alpha = 0.05$. The analysis was performed using SPSS v. 23.

## Ethical considerations

The study was approved by the National Medical Ethics Committee of the Republic of Slovenia on January 1, 2015 (No. 43/01/15 and ref. no. 0120-68/2018/8). The research presented is observational; all data of included healthcare (HEC) providers were collected in such a way that the anonymity of included HEC providers was fully ensured, including in their department. Therefore, the National Medical Ethics Committee of the Republic of Slovenia stated that for participation in the research and the use of necessary data, no written informed consent was needed for their inclusion.

## Results

The questionnaire was sent to the following 14 hospitals in Slovenia (the response rates in percentages (%) and the number of sent questionnaires for each hospital are in brackets). The three tertiary level University Hospitals were: Ljubljana (52% out of 444), Maribor (44% out of 141), and Golnik (62% out of 21). The eleven secondary level general hospitals were: Topolšica (80% out of 10), Jesenice (26% out of 34), Izola (28% out of 40), Nova Gorica (41% out of 39), Novo Mesto (41% out of 48), Brežice (68% out of 19), Ptuj (80% out of 25), Murska Sobota (85% out of 40), Slovenj Gradec (38% out 37), Celje (28% out of 86), and Trbovlje (57% out of 14). The response rates in the secondary and tertiary level institutions were 45% and 51%, respectively. The final sample size was n = 485.

**Table 1. Sample characteristics of HealthCare Professionals (HCPs) (n = 485).**

|  | n (%) |
|---|---|
| **GENDER** |  |
| Male | 100 (20.6) |
| Female | 385 (79.4) |
| **Mean age (SD; n = 481)** | 40.9 (10.6) |
| **Mean years of employment (SD; n = 484)** | 18.8 (11.4) |
| **Mean years of employment in current hospital (SD; n = 484)** | 16.6 (11.1) |
| **Religion (n = 480)** |  |
| Religious | 364 (75.8) |
| Not religious | 56 (11.7) |
| **TYPE OF INSTITUTION (n = 471)** |  |
| Secondary level | 177 (36.5) |
| Tertiary level | 308 (63.5) |
| **PROFILES OF HCPs** |  |
| Physicians | 76 (15.7) |
| Nurses | 320 (**66**) |
| Other HCPs | 89 (**18.4**) |
| **HOSPITAL UNIT (n = 484)** |  |
| Reception clinic | 21 (4.3) |
| Clinic | 75 (15.5) |
| Emergency wards | 24 (5) |
| Hospital wards | 184 (**38**) |
| Intensive care units | 50 (10.3) |
| Diagnostic wards | 36 (7.4) |
| Surgical wards | 53 (11) |
| Elsewhere | 41 (8.5) |

HCPs–healthcare professionals. Other HCPs—laboratory technicians and engineers, radiological engineers, clinical psychologists, nurse assistants, biochemical technicians and engineers, pharmacists, social workers, physiotherapists, respiratory therapists, speech therapists, hygiene technicians, and psychologists.

## Characteristics of participating HCPs

The demographic characteristics of HCPs participating in the study are presented in Table 1. The sample consisted of 385 (79.4%) females. The mean (standard deviation) age, employment period and years of employment in the current hospital were 40.9 (10.6), 18.8 (11.4) and 16.6 (11.1), respectively. The sample consisted of 76 (15.7%) physicians, 320 (66%) nurses, and 89 (18.4%) other HCPs.

The proportions of the profiles of HCPs included in the sample statistically significantly differ from the proportions in the population (p = 0.036). In our sample, physicians were underrepresented (23 more physicians were expected in the sample of the given size) whereas nurses were overrepresented (26 fewer nurses were expected in the sample of the given size). There were 211 (44.8%) HCPs from secondary level institutions and 260 (55.2%) HCPs from tertiary level institutions included in the sample. The distribution of HCPs by type of institution did not differ statistically significantly from that in the general population (p = 0.183).

Out of 483 HCPs who answered the questions on ethical dilemmas, 340 (70.4%; 95% CI: 66.7–74.5%) (very) frequently encountered ethical dilemmas during their work (Table 2). The percentages of physicians, nurses, and other HCPs (very) often encountering ethical dilemmas

**Table 2. The percentages of (very) frequently encountered ethical dilemmas and the association between type of HCPs and ethical dilemmas.**

| | Physicians (n = 76) | | Nurses (n = 320) | | Other HCPs (n = 89) | | All HCPs (n = 485) | | Physicians | | Nurses | |
|---|---|---|---|---|---|---|---|---|---|---|---|---|
| | n (%) | R | n (%) | R | n (%) | R | n (%) | R | OR (95 CI) | p-value | OR (95 CI) | p-value |
| **Ethical dilemma** | **69 (90.8)** | **76** | **216 (67.7)** | **319** | **55 (62.5)** | **88** | **340 (70.4)** | **483** | **5.8 (2.3; 14.5)** | **< 0.001** | **1.1 (0.7; 1.9)** | **0.601** |
| **Waiting periods for diagnostics or therapeutic treatment** | 53 (69.7) | 76 | 113 (36.2) | 312 | 21 (26.6) | 79 | 187 (40) | 467 | 6.8 (3.3; 14.1) | < 0.001 | 1.7 (1; 3) | 0.073 |
| **Suboptimal working conditions due to poor interpersonal relations on the ward** | 40 (52.6) | 76 | 94 (29.9) | 314 | 18 (21.7) | 83 | 152 (32.1) | 473 | 4.2 (2; 8.6) | < 0.001 | 1.6 (0.9; 2.9) | 0.12 |
| **Preserving patients' dignity** | 23 (30.7) | 75 | 95 (30.4) | 313 | 23 (28.8) | 80 | 141 (30.1) | 468 | 1.2 (0.6; 2.6) | 0.614 | 1.1 (0.6; 2) | 0.782 |
| **Relations between HCPs and patients (or their relatives)** | 25 (32.9) | 76 | 91 (29.2) | 312 | 20 (25.3) | 79 | 136 (29.1) | 467 | 1.5 (0.7; 3.1) | 0.272 | 1.2 (0.7; 2.1) | 0.594 |
| **Recognising a patient's best interests** | 29 (38.7) | 75 | 73 (23.5) | 310 | 25 (31.3) | 80 | 127 (27.3) | 465 | 1.5 (0.8; 3.1) | 0.242 | 0.7 (0.4; 1.2) | 0.189 |
| **Protection of patient information** | 15 (19.7) | 76 | 85 (27.1) | 314 | 25 (30.9) | 81 | 125 (26.5) | 471 | 0.5 (0.2; 1.1) | 0.081 | 0.8 (0.5; 1.5) | 0.529 |
| **End-of-life treatment withdrawal** | 36 (49.3) | 73 | 78 (25.2) | 309 | 5 (6.3) | 79 | 119 (25.8) | 461 | 15.8 (5.6; 44.6) | < 0.001 | 5.3 (2; 13.7) | < 0.001 |
| **New modes of treatment and diagnostic procedures** | 20 (26.3) | 76 | 67 (21.5) | 311 | 10 (12.7) | 79 | 97 (20.8) | 466 | 2.5 (1.04; 5.8) | **0.039** | 2 (1; 4.2) | 0.064 |
| **Allocation of limited resources** | 28 (37.3) | 75 | 53 (17.2) | 309 | 14 (17.3) | 81 | 95 (20.4) | 465 | 2.9 (1.4; 6.2) | **0.006** | 1 (0.5; 1.9) | 0.917 |
| **Lack of response to adverse events in patient management** | 25 (32.9) | 76 | 50 (16.2) | 309 | 8 (9.9) | 81 | 83 (17.8) | 466 | 4.6 (1.9; 11.4) | < 0.001 | 1.7 (0.8; 3.9) | 0.18 |
| **Insufficient availability of palliative care** | 20 (27) | 74 | 59 (19) | 311 | 2 (2.6) | 78 | 81 (17.5) | 463 | 18.1 (3.9; 83.7) | < 0.001 | 10.4 (2.4; 45) | **0.002** |
| **A patient's consent to undergo a diagnostic or therapeutic procedure** | 14 (18.7) | 75 | 64 (20.4) | 313 | 8 (10.1) | 79 | 86 (18.4) | 467 | 2.2 (0.8; 5.6) | 0.113 | 2.3 (1.1; 5.2) | **0.037** |
| **Disagreement with an individual's professional work** | 18 (23.7) | 76 | 55 (17.7) | 310 | 7 (8.6) | 81 | 80 (17.1) | 467 | 3.5 (1.3; 9.3) | **0.011** | 2.5 (1.1; 6) | **0.034** |
| **Learning on patients** | 10 (13.2) | 76 | 51 (16.3) | 312 | 14 (17.7) | 79 | 75 (16.1) | 467 | 0.7 (0.3; 1.8) | 0.503 | 0.9 (0.4; 1.8) | 0.728 |
| **A patient's right to refuse treatment** | 8 (10.5) | 76 | 48 (15.4) | 311 | 2 (2.6) | 77 | 58 (12.5) | 464 | 4.4 (0.9; 21.7) | 0.07 | 7.2 (1.7; 30.7) | **0.008** |
| **Social inequality or withdrawal of basic healthcare insurance** | 12 (15.8) | 76 | 34 (10.9) | 311 | 2 (2.5) | 79 | 48 (10.3) | 466 | 7 (1.5; 32.9) | **0.013** | 4.5 (1.1; 19.5) | **0.042** |
| **Involuntary hospitalisation** | 8 (10.7) | 75 | 33 (10.6) | 312 | 2 (2.6) | 77 | 43 (9.3) | 464 | 4.4 (0.9; 21.6) | 0.07 | 4.5 (1; 19.4) | **0.043** |
| **Biomedical research** | 4 (5.3) | 75 | 21 (6.8) | 308 | 5 (6.5) | 77 | 30 (6.5) | 460 | 0.8 (0.2; 3) | 0.687 | 1.1 (0.4; 3) | 0.919 |
| **Organ transplantation** | 4 (5.5) | 73 | 20 (6.5) | 306 | 2 (2.5) | 79 | 26 (5.7) | 458 | 2.2 (0.4; 12.6) | 0.374 | 3 (0.7; 13.4) | 0.145 |
| **Refusal of Vaccines** | 4 (5.4) | 74 | 19 (6.1) | 312 | 3 (3.8) | 78 | 26 (5.6) | 464 | 1.3 (0.3; 6.1) | 0.753 | 1.6 (0.4; 5.6) | 0.482 |

Results of univariate mixed-effect logistic regression with other HCPs as the reference category. HCPs–healthcare professionals, R–number of respondents, OR–odds ratio adjusted for hospital, CI–confidence interval.

were 90.8%, 67.7%, and 62.5%, respectively. Logistic regression showed that physicians had ~six-times higher odds of facing an ethical dilemma compared to other HCPs (OR = 5.8; 95% CI: 2.3–14.5; Table 2). The odds of nurses frequently encountering an ethical dilemma were comparable to those of other HCPs (p = 0.601). Of all the ethical dilemmas, the one most frequently encountered by the largest proportion of HCPs was waiting periods for diagnostics or therapeutic treatment (40%), followed by suboptimal working conditions due to poor interpersonal relations on the ward (32.1%), preserving patients' dignity (30.1%), and relations between healthcare professionals and patients (or their legal guardians) (29.1%). Frequent ethical dilemmas recognised by the largest percentages of physicians, nurses, and other HCPs are presented in Table 2.

Compared to other HCPs, physicians had higher odds of encountering ethical dilemmas regarding new modes of treatment and diagnostic procedures, end-of-life treatment withdrawal, suboptimal working conditions due to poor interpersonal relations on the ward,

disagreement with an individual's professional work, lack of response to adverse events in patient management, social inequality or withdrawal of patients' rights to basic healthcare insurance, allocation of limited resources, insufficient availability of palliative care, and waiting periods for diagnostics or treatment (Table 2).

Compared to other HCPs, nurses had statistically significantly higher odds of encountering ethical dilemmas regarding end-of-life treatment withdrawal, disagreement with an individual's professional work, social inequality or withdrawal of patients' rights to basic healthcare insurance, and a patient's consent to undergo a diagnostic or therapeutic procedure (Table 2).

## Ethical dilemmas among HCPs working in secondary and tertiary level institutions

Physicians and nurses working in secondary level institutions more frequently encountered ethical dilemmas regarding preserving patients' dignity, protection of patient information, and relations between HCPs and patients compared to their colleagues working in tertiary level institutions (Fig 1). In addition, physicians working at the secondary level more often encountered ethical dilemmas pertaining to a patient's consent to undergo a diagnostic or therapeutic procedure compared to those working in tertiary level institutions. Nurses working in a secondary level institution were more often faced with ethical dilemmas of recognising the patient's best interest compared to nurses working in tertiary level institutions. Physicians and nurses working in secondary level institutions had higher odds of more frequently encountering the above-mentioned ethical dilemmas than physicians and nurses working in tertiary level institutions (see S1–S3 Tables).

## Whom do HCPs consult when confronted with an ethical dilemma?

Table 3 shows the results concerning whom HCPs consult when faced with an ethical dilemma in their work. Most frequently they discussed it with colleagues (94.2%) and/or with the head of the department (55.2%). Physicians most commonly discussed ethical dilemmas with colleagues (90.8%) or with the head of the department (75%), or they called a medical council meeting

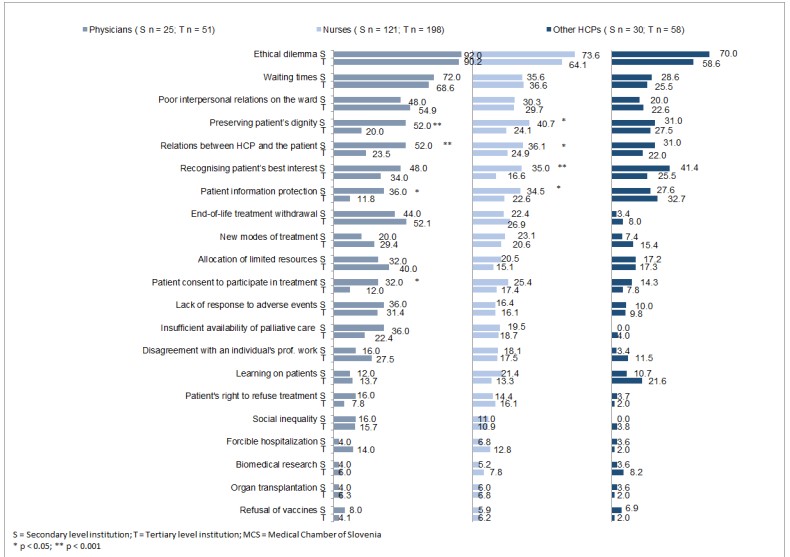

**Fig 1. Frequent ethical dilemmas by the type of institution.** S–secondary level institution, T–Tertiary Level Institution, HCPs–Healthcare Professionals, MCS–Medical Chamber of Slovenia.

(47.4%). Nurses and other HCPs most commonly discussed ethical dilemmas with colleagues (96.3%, 89.5%, respectively) and with the head of the department (48.8%, 61.6%, respectively). Physicians had higher odds (OR = 96.9; 95% CI: 12.4–757.2) of calling a medical council meeting or of consulting with an HEC (OR = 17.8; 95% CI: 3.9–82.3) than other HCPs.

Compared to other HCPs, nurses had lower odds (OR = 0.6; 95% CI: 0.3–0.9) of discussing ethical dilemmas with the head of the department or of deciding alone (OR = 0.3; 95% CI: 0.2–0.7), but higher odds of discussing ethical dilemmas with colleagues (OR = 3.2; 95% CI: 1.2–8.1).

Fig 2 shows the results of reactions to situations involving ethical dilemmas for each group of HCPs working in secondary and tertiary level institutions.

Physicians in secondary level institutions had higher odds (OR = 5.1; 95% CI: 1.1–22.3) of resolving ethical dilemmas in the HCP's family circle and lower odds (OR = 0.1; 95% CI: 0.01–0.6) of discussing the dilemma with an HEC.

Nurses in secondary level institutions had higher odds (OR = 2.2; 95% CI: 1.4–3.5) of discussing ethical dilemmas with the head of the department and of resolving the dilemma through mediation (OR = 4; 95% CI: 1–15.6) than nurses working in tertiary level institutions.

Other HCPs working in secondary level institutions had higher odds (OR = 3.4; 95% CI: 1.2–9.7) of discussing the ethical dilemma with the head of the department and of deciding alone (OR = 11; 95% CI: 2.8–43.9) compared to those working in tertiary level institutions.

In the investigated groups of HCPs, the highest proportion of HCPs who were unaware of standard procedures for solving ethical dilemmas was found among physicians (30.7%) compared to nurses (13.2%) and other HCPs (6.7%). From those who answered "yes", "to being aware" or "no" to being unaware, the odds of answering "yes" were lower for physicians (OR = 0.2; 95% CI: 0.1–0.6) compared to other HCPs (Table 4).

The odds of not discussing ethical dilemmas were higher among physicians than other HCPs (OR = 2.9; 95% CI: 1.2–7.1), but not for nurses compared to other HCPs (p = 0.453).

Considering all HCPs (n = 483), the most common ways of learning about medical ethics were in the educational system or university (24%), learning from more experienced peers (21.3%), and learning at workshops (20.3%). No differences were found between HCPs. The

**Table 3. Association between profiles of HCPs and ways of dealing with ethical dilemmas.**

| | Physicians (n = 76) | Nurses (n = 320) | Other HCPs (n = 86) | All HCPs (n = 482) | Physicians | | Nurses | |
|---|---|---|---|---|---|---|---|---|
| | n (%) | n (%) | n (%) | n (%) | OR (95 CI) | p-value | OR (95 CI) | p-value |
| **Discuss with co-workers** | 69 (90.8) | 308 (96.3) | 77 (89.5) | 454 (94.2) | 1.4 (0.5; 4.2) | 0.556 | 3.2 (1.2; 8.1) | **0.017** |
| **Discuss with head of the department** | 57 (75) | 156 (48.8) | 53 (61.6) | 266 (55.2) | 2 (1; 4) | 0.057 | 0.6 (0.3; 0.9) | **0.029** |
| **Decide on my own** | 13 (17.1) | 24 (7.5) | 14 (16.3) | 51 (10.6) | 1.1 (0.4; 2.5) | 0.902 | 0.3 (0.2; 0.7) | **0.006** |
| **Call a medical council meeting** | 36 (47.4) | 7 (2.2) | 1 (1.2) | 44 (9.1) | 96.9 (12.4; 757.2) | **< 0.001** | 2 (0.2; 16.6) | 0.534 |
| **Consult the hospital ethics committee** | 19 (25) | 10 (3.1) | 2 (2.3) | 31 (6.4) | 17.8 (3.9; 82.3) | **< 0.001** | 1.6 (0.3; 7.5) | 0.561 |
| **Resolve the issue in family circles** | 9 (11.8) | 11 (3.4) | 4 (4.7) | 24 (5) | 2.6 (0.7; 9.5) | 0.146 | 0.5 (0.2; 1.8) | 0.313 |
| **Consult with patient's legal representative** | 3 (3.9) | 9 (2.8) | 1 (1.2) | 13 (2.7) | 3.3 (0.3; 33.2) | 0.32 | 2 (0.2; 16.7) | 0.532 |
| **Resolve the issue with mediation** | 3 (3.9) | 10 (3.1) | 0 (0) | 13 (2.7) | | | | |
| **Consult the hospital chaplain** | 1 (1.3) | 5 (1.6) | 0 (0) | 6 (1.2) | | | | |
| **Consult the national ethics committee** | 4 (5.3) | 0 (0) | 0 (0) | 4 (0.8) | | | | |
| **Consult the Human Rights Ombudsman** | 2 (2.6) | 1 (0.3) | 1 (1.2) | 4 (0.8) | 2 (0.2; 24.5) | 0.577 | 0.2 (0.01; 4.2) | 0.333 |
| **Consult the Committee for Legal and Ethical Issues of the Medical Chamber of Slovenia** | 0 (0) | 1 (0.3) | 1 (1.2) | 2 (0.4) | | | 0.3 (0; 4.4) | 0.358 |
| **Contact the media** | 0 (0) | 0 (0) | 1 (1.2) | 1 (0.2) | | | | |

Results of univariate mixed-effect logistic regression with other HCPs as the reference category. HCPs–healthcare professionals, OR–odds ratio adjusted for hospital, CI–confidence interval.

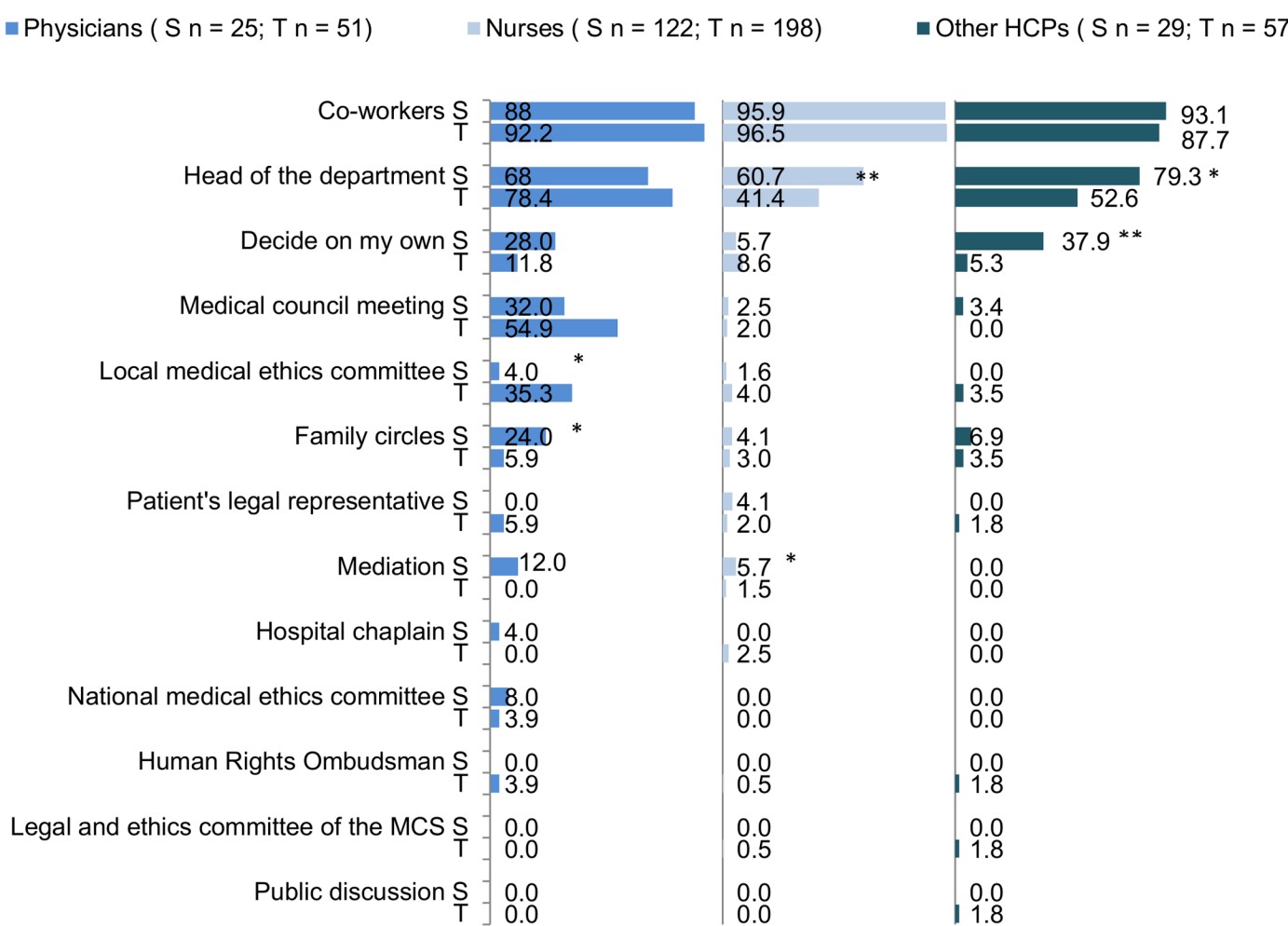

**Fig 2. Ways of dealing with ethical dilemmas by type of institution.** S–secondary level institution, T–tertiary level institution, HCPs–healthcare professionals, MCS–Medical Chamber of Slovenia.

majority (88.7%) of HCPs believed that regular formal education about ethics in medicine is required for hospital staff. We found no differences between working profiles.

The role of HECs was perceived to be important by 84.6% of HCPs, and specifically by 79.7% of physicians, 84.9% of nurses, and 87.6% of other HCPs. No statistically significant differences in opinion on the importance of HECs were found between the groups of HCPs (Table 5). About half (51.4%) of HCPs were aware of the existence of an ethics committee in their hospital. There were no statistically significant differences in the proportions of physicians, nurses, and other HCPs who were aware of ethics committees in their hospital (Table 5). More than a third (36.5%) of all HCPs knew they could consult the HEC. There were no statistically significant differences in the proportions of physicians, nurses, and other HCPs in terms of their familiarity with the option to consult HECs (Table 5).

**Table 4. Association between profiles of HCPs and standard procedures for solving ethical dilemmas or the presence of an undiscussed ethical dilemma.**

| | Physicians | Nurses | Other HCPs | All HCPs | Physicians | | Nurses | |
|---|---|---|---|---|---|---|---|---|
| | | | | | OR (95 CI) | p-value | OR (95 CI) | p-value |
| **Standard procedures for resolving ethical dilemmas** | | | | | | | | |
| Yes, to being aware of | 21 (28) | 106 (33.3) | 26 (29.2) | 153 (31.7) | 0.2 (0.1; 0.6) | **0.004** | 0.6 (0.2; 1.6) | 0.288 |
| No, to being unaware of | 23 (30.7) | 42 (13.2) | 6 (6.7) | 71 (14.7) | 1 | | 1 | |
| Do not know | 31 (41.3) | 170 (53.5) | 57 (64) | 258 (53.5) | - | - | - | - |
| **Undiscussed ethical dilemma** | | | | | | | | |
| Yes | 25 (32.9) | 63 (19.7) | 12 (13.6) | 100 (20.7) | 2.9 (1.2; 7.1) | **0.019** | 1.3 (0.6; 2.9) | 0.453 |
| No | 36 (47.4) | 172 (53.9) | 45 (51.1) | 253 (52.4) | 1 | | 1 | |
| Do not know | 15 (19.7) | 84 (26.3) | 31 (35.2) | 130 (26.9) | - | - | - | - |

Results of univariate logistic regression with other HCPs as the reference category. HCPs–healthcare professionals, OR–odds ratio adjusted for hospital, CI–confidence interval.

The number of consultations with HECs in 2014 was 0–2 for physicians, 0–4 for nurses, and 0–1 for other HCPs. Physicians had higher odds (OR = 5.3; 95% CI: 1.6–17.6) of having consulted with an HEC in 2014 compared to other HCPs (p = 0.007).

None of the physicians or nurses and one of the other HCPs working in a secondary level hospital had consulted with an HEC in 2014. Among the physicians working in a tertiary level institution, 15 (33.3%) had consulted HECs in 2014. The corresponding number of nurses was 12 (6.9%), while three (6.4%) of the other HCPs had consulted the HEC.

All HCPs stated that the most important role of HECs was healthcare staff education (47.1%), followed by improving communication (41.1%), and review of difficult cases (35.1%). Physicians, compared to other HCPs, gave more importance to review of difficult cases (OR = 2; 95% CI: 1.04–3.8), preparation of guidelines or protocols (OR = 2.8; 95% CI: 1.3–5.8), legal protection of physicians in the decision-making process (OR = 18.4; 95% CI: 4.1–82.8), and allocation of limited resources (OR = 13.5; 95% CI: 1.6–111.6). On the other hand, they placed less importance on improving communication (OR = 0.3; 95% CI: 0.2–0.6) or conflict

**Table 5. Relationship between profiles of HCPs and the perceived importance of the HEC, awareness of its existence, or awareness of the option to consult the HEC.**

| | Physicians | Nurses | Other HCPs | All HCPs | Physicians | | Nurses | |
|---|---|---|---|---|---|---|---|---|
| | | | | | OR (95% CI) | p-value | OR (95% CI) | p-value |
| **Importance of the role of the medical ethics committee** | | | | | | | | |
| Weaker agreement | 15 (20.3) | 48 (15.1) | 11 (12.4) | 74 (15.4) | | | | |
| Stronger agreement | 59 (79.7) | 270 (84.9) | 78 (87.6) | 407 (84.6) | 0.6 (0.3; 1.5) | 0.28 | 0.8 (0.4; 1.7) | 0.606 |
| **Existence of ethics committee** | | | | | | | | |
| Yes | 42 (56) | 165 (51.7) | 40 (46) | 247 (51.4) | 0.4 (0.1; 1.9) | 0.23 | 1.1 (0.3; 3.4) | 0.908 |
| No | 9 (12) | 19 (6) | 4 (4.6) | 202 (42) | | | | |
| Do not know | 24 (32) | 135 (42.3) | 43 (49.4) | 32 (6.7) | - | - | - | - |
| **Option of consultations for staff*** | | | | | | | | |
| Yes | 27 (52.9) | 84 (36.2) | 14 (23.7) | 125 (36.5) | 2.1 (0.5; 8.9) | 0.313 | 1.7 (0.5; 5.7) | 0.357 |
| No | 5 (9.8) | 25 (10.8) | 5 (8.5) | 182 (53.2) | | | | |
| Do not know | 19 (37.3) | 123 (53) | 40 (67.8) | 35 (10.2) | - | - | - | - |

Results of univariate logistic regression with other HCPs as the reference category. HCPs–healthcare professionals, OR–odds ratio adjusted for hospital, CI–confidence interval, *–number of respondents = 342; physicians n = 51, nurses n = 232, and other HCPs n = 5.

**Table 6. Association between profiles of HCPs and the role of HECs.**

| | Physicians (n = 76) | Nurses (n = 320) | Other HCPs (n = 89) | All HCPs (n = 485) | Physicians | | Nurses | |
|---|---|---|---|---|---|---|---|---|
| | | | | | OR (95% CI) | p-value | OR (95% CI) | p-value |
| Education of HCPs | 31 (40.8) | 151 (47.5) | 45 (51.1) | 227 (47.1) | 0.6 (0.3; 1.2) | 0.148 | 0.8 (0.5; 1.4) | 0.493 |
| Improving communication | 16 (21.1) | 139 (43.7) | 43 (48.9) | 198 (41.1) | 0.3 (0.2; 0.6) | **0.001** | 0.9 (0.5; 1.4) | 0.586 |
| Review of difficult cases | 37 (48.7) | 103 (32.4) | 29 (33) | 169 (35.1) | 2 (1.04; 3.8) | **0.037** | 1 (0.6; 1.6) | 0.932 |
| Conflict resolution | 7 (9.2) | 103 (32.4) | 27 (30.7) | 137 (28.4) | 0.2 (0.1; 0.6) | **0.003** | 1.1 (0.7; 1.9) | 0.669 |
| Moral support for HCPs | 14 (18.4) | 96 (30.2) | 18 (20.5) | 128 (26.6) | 0.9 (0.4; 2.1) | 0.86 | 1.6 (0.9; 3) | 0.109 |
| Ethical consultations for employees | 20 (26.3) | 69 (21.7) | 22 (25) | 111 (23) | 1 (0.5; 2.1) | 0.961 | 0.9 (0.5; 1.5) | 0.621 |
| Improving the quality of healthcare | 13 (17.1) | 55 (17.3) | 23 (26.1) | 91 (18.9) | 0.6 (0.3; 1.3) | 0.17 | 0.5 (0.3; 0.9) | **0.023** |
| Developing guidelines or protocols | 27 (35.5) | 48 (15.1) | 15 (17) | 90 (18.7) | 2.8 (1.3; 5.8) | **0.008** | 0.9 (0.5; 1.8) | 0.788 |
| Ethical consultations in the wards (at the bedside) | 10 (13.2) | 40 (12.6) | 9 (10.2) | 59 (12.2) | 1.2 (0.4; 3.1) | 0.764 | 1.2 (0.5; 2.6) | 0.67 |
| Support of patients in giving them a stronger voice in decision-making | 5 (6.6) | 45 (14.2) | 12 (13.6) | 62 (12.9) | 0.5 (0.2; 1.5) | 0.206 | 1.1 (0.6; 2.3) | 0.746 |
| Legal protection of physicians in decision-making | 23 (30.3) | 26 (8.2) | 2 (2.3) | 51 (10.6) | 18.4 (4.1; 82.8) | **< 0.001** | 3.7 (0.9; 16.3) | 0.079 |
| Assessment on introduction of novel treatment modes | 9 (11.8) | 23 (7.2) | 12 (13.6) | 44 (9.1) | 0.8 (0.3; 2.1) | 0.639 | 0.5 (0.2; 1.1) | 0.088 |
| Counselling hospital management staff | 3 (3.9) | 23 (7.2) | 2 (2.3) | 28 (5.8) | 1.7 (0.3; 10.7) | 0.567 | 3.7 (0.8; 16.3) | 0.082 |
| Allocation of limited resources | 9 (11.8) | 12 (3.8) | 1 (1.1) | 22 (4.6) | 13.5 (1.6; 111.6) | **0.016** | 3.7 (0.5; 29.5) | 0.212 |
| Other | 0 (0) | 2 (0.6) | 0 (0) | 2 (0.4) | - | - | - | - |

Results of univariate logistic regression with other HCPs as the reference category. HCPs–healthcare professionals, OR–odds ratio adjusted for hospital, CI–confidence interval.

resolution (OR = 0.2; 95% CI: 0.1–0.6). Compared to other HCPs, nurses attributed less importance to the improvement in the quality of healthcare (OR = 0.5; 95% CI: 0.3–0.9) (Table 6).

## Discussion

This is the first national survey among healthcare professionals (HCPs; physicians, nurses, and other HCP profiles) from 14 hospitals (secondary and tertiary level institutions) examining ethical dilemmas confronting HCPs in everyday professional practice, how they solve them, how they use hospital ethics committees (HECs), and their opinions about resolving their ethical dilemmas. This study revealed several important findings.

Firstly, all HCPs who participated in the study (very) often encountered ethical dilemmas (70.4%). However, among physicians, the proportion rose to 90.8% whereas the proportions of nurses and other HCPs who encountered ethical dilemmas were 67.7% and 62.5%, respectively. This resulted in physicians having six times higher odds of facing ethical dilemmas compared to other HCPs. A possible reason for this result is the fact that the Physicians Practitioners Act endorses physicians as the sole responsible persons for medical activities, whereas the legislature failed to regulate the professional activity of nurses and other HCPs [40]. No statistically significant differences were found between nurses and other HCPs, or between physicians working in either secondary or tertiary level institutions. The proportion of ethical dilemmas among Slovenian HCPs is comparable with the findings of other studies that showed that between 60% and 90% of HCPs encountered various ethical dilemmas in their work [13, 32, 41–43]. Our findings concur with those of several international studies,

which reported that not only physicians but also nurses and other HCPs may encounter different ethical dilemmas [42, 44–46, 47]. The demographic characteristics of our HCPs showed that the majority of our participants were female (79.5%). In our study sample, physicians were underrepresented, and nurses overrepresented, whereas the distribution among secondary and tertiary level institutions did not statistically significantly differ from that in the general population. In a previous study performed on a sample from a tertiary level hospital [37], we observed that 60% of HCPs encountered an ethical dilemma, whereas in our study a higher percentage of respondents (70.4%) frequently encountered ethical dilemmas. We collected data from all types of hospital workplaces, that is, from reception clinics, clinics, emergency wards, hospital wards, intensive care units, diagnostic wards, and surgical wards, as well as from other workplaces not previously mentioned. This gave us good insight into a whole array of ethical dilemmas that may confront different profiles of HCPs. Other studies mostly targeted only one or two different profiles of HCPs. DuVal et al. developed a questionnaire based on a review of ethics consultations and later included a cognitive method to prepare the questionnaire. The same questionnaire was then also used by Hurst and Sorta-Bilajac to study ethical issues in clinical practice [13, 38–43]. DuVal et al. found that among surveyed American physicians (general internists, oncologists, and critical care specialists), general internists mostly reported dilemmas regarding end-of-life decision-making, patient autonomy, justice, and conflict resolution [13]. End-of-life decisions and patient autonomy were often referred for consultation, while dilemmas about justice, such as lack of insurance or limited resources, were rarely referred. Physicians who are more knowledgeable and experienced in ethics are significantly more likely to request an ethics consultation. The study of duVal et al. also revealed that 41% of physicians expressed some hesitation in requesting ethics consultations. This result concurs with ours as we found that 35.3% of physicians working in tertiary level hospitals would discuss an ethical dilemma with the hospital ethics committee compared to only 4% of physicians in secondary level hospitals (p = 0.015). No differences were found for nurses and other HCPs. Therefore, we ~~can~~ conclude that not only physicians in secondary level hospitals but also nurses and other HCPs in both secondary and tertiary level hospitals are probably insufficiently aware of the existence of hospital ethics committees, and, therefore, they do not consult them when faced with ethical dilemmas. More education is probably needed to address this issue.

Hurst et al. performed a study among general internists in Norway, Switzerland, Italy, and the UK, and found that uncertain or impaired decision-making capacity, disagreement among caregivers and limitation of treatment at the end-of-life were their most frequent ethical dilemmas [43]. The third study employing the same questionnaire was performed among Croatian physicians and nurses at the University Hospital Rijeka and this yielded similar result [41–43]. The results from those three studies show that some of the issues raised are remarkably similar or identical to those in our study, although there are important differences. Among the most important ethical dilemmas in both their studies and ours, were ethical dilemmas concerning end-of-life decisions and relationships between HCPs, but the two most important dilemmas in our study, which were not included in the other studies, were "waiting periods for diagnostics or therapeutic treatment" and "interpersonal relationships on the ward" which were reported by 69.7% and 52.6% of physicians and much less often by nurses and other HCPs. Because euthanasia and assisted suicide are illegal in Slovenia, we did not pose such questions in our questionnaire. If we assume that waiting periods in Slovenia are probably not only due to a shortage of healthcare personnel but also due to a centrally planned model of healthcare still governed by a single national medical insurance company, it becomes clear that lack of financial support for all diagnostics, operations, and treatments also plays a major role. Hospitals are, therefore, permitted to provide only planned and approved healthcare diagnostics and

procedures and nothing more, which is the source of great ethical dilemmas for Slovenian HCPs.

HCPs encounter different types of ethical dilemmas to varying extents depending on their HCP profile. While waiting periods, interpersonal relationships on the ward, and end-of-life treatment are the most common dilemmas among physicians, care for a patient's dignity, besides the first two dilemmas listed for physicians, are most often encountered by nurses. Among other HCPs, recognising the patient's best interests, protection of patient information, and care for the patient's dignity are the three most important ethical dilemmas. Possible reasons for their answer are: a.) other HCPs have access to a large amount of data and information when they take care of the patients specifically to their profession, b.) that nobody teaches them how to manage and interpret sensitive personal information they encounter.

However, no differences in any of the ethical dilemmas were found between secondary and tertiary level institutions for other HCPs. Waiting periods and interpersonal relationships on the ward did not differ significantly among physicians and nurses in secondary and tertiary levels of institutions for all HCPs, which clearly shows that this is a major problem in Slovenian hospitals. The lack of professionalism among physicians is one of the great problems leading to poor interpersonal relationships on the ward [48]. Among Bulgarian physicians, predominant dilemmas included relationships with patients and relatives (76.8%) and teamwork (67.6%), followed by end-of-life issues (31.5%) [49]. Nurses and other HCPs, who have the primary responsibility of caring for the patient, are somewhat more alert to the dilemmas associated with patients' well-being and dignity. Similarly, Norberg et al. observed that nurses are more emotionally involved in ethical dilemmas concerning patient care compared to physicians [45–50].

Higher proportions of physicians and nurses in secondary level institutions experienced ethical dilemmas regarding care for a patient's dignity, the relationship between HCPs and patients, and protection of patient information. One of the reasons for higher proportions among HCPs working in secondary level institutions is because Slovenia is a relatively small country in terms of its size and number of inhabitants. Secondary level hospitals are found in small towns with close-knit communities where protected information may leak quickly. In addition, the protection of personal information became one of the most pressing security concerns for record keepers, especially after the introduction of the European General Data Protection Regulation (GDPR) in 2014 and the Personal Data Protection Act into Slovenian legislation in 2007 [51–53].

Five types of ethical dilemmas were universally given the lowest priority by Slovenian HCPs: social inequality, involuntary hospitalisation, biomedical research, organ transplantation, and vaccination refusal. Concerning the latter two dilemmas, organ transplantation is legally well regulated and performed in only one tertiary centre in the whole country; the Institute for Transplantation of Organs and Tissues of the Republic of Slovenia is a member of the Eurotransplant organisation [54].

Despite some tendency for parents to refuse to vaccinate their children, the percentage of vaccinated children nevertheless remains high in Slovenia and, therefore, this is not of concern for HCPs in secondary and tertiary level institutions, but rather in primary healthcare institutions, which do not have their own ethics committees.

In all our HCP groups, a high percentage (94.2%) of respondents resolved ethical dilemmas in discussion with their colleagues, with no differences being found between secondary and tertiary level institutions. The second most frequent approach in all three HCP groups was to discuss the ethical dilemma with the head of the department (55.2%); nurses and other HCPs working in secondary level institutions used this option more frequently than their peers working in tertiary level institutions. The medical council meeting was considered by

physicians to be a very important platform for discussing ethical dilemmas, with no differences found between secondary and tertiary level institutions. HECs were consulted more often in tertiary (35.3%) compared to secondary level institutions, where HECs are very rarely consulted (4.0%). Physicians in secondary level institutions had significantly higher odds of discussing their ethical dilemmas within their family circle compared to physicians working in tertiary level institutions (24.0% vs 5.9%; OR = 5.1), whereas nurses in secondary level institutions preferred mediation compared to nurses in tertiary level institutions (5.7% vs 1.5%; OR = 4.0). Other HCPs in secondary level institutions had a higher odds ratio of deciding alone than their counterparts in tertiary level institutions (OR = 11). Discussing ethical dilemmas with co-workers proved to be the most frequent strategy chosen by the respondents in our study. In their study on how HCPs solve ethical dilemmas, Moeller et al. reviewed 100 cases of ethical consultations. They found that the reasons for consultations could be divided into one of eight general categories: conflict over withholding treatment, conflict over withdrawing treatment, futility issues, and the decisional capacity of the patient in question, wishes of the patient unknown, patient non-compliant with the medical regimen, issues with DNR status, and other [55].

In our previous nationwide study on experiences of intensivists in intensive care units of end-of-life attitudes and how to proceed when faced with ethical dilemmas, only 60% of the study participants (intensivists) knew how to proceed when facing an ethical dilemma, while 23% of all the participants had previously consulted an HEC. Furthermore, 42% of the respondents knew the name of the head of the HEC in their institution, whereas 17% reported that there was no HEC in their institution [56].

In our study, it would be interesting to establish with whom the physicians consulted in cases of ethical dilemmas concerning end-of-life decision-making. The most common way of dealing with ethical dilemmas is to consult colleagues and/or the head of the department. In secondary level institutions, physicians are significantly less likely to consult HECs compared to those working in tertiary level institutions (OR = 0.1). Nurses in secondary level institutions preferred mediation compared to those working in tertiary level institutions (OR = 4). As regards consulting HECs, no differences were found between all three HCP groups. In the study on Croatian physicians and nurses, 12% of physicians and only 3% of nurses consulted the HEC [41], which was much lower than in comparable tertiary institutions in Slovenia.

Our results revealed that physicians (30.7%) were more unaware of standard procedures for solving ethical dilemmas than nurses (13.2%) and other HCPs (6.7%). A very high proportion of physicians (41.3%), nurses (53.3%), and other HCPs (64.0%) were not aware of existing standard procedures for solving ethical dilemmas. Together, this results in a very high proportion of all HCPs that do not know and/or are unaware of standard procedures, even though a third of physicians compared to 19.7% of nurses and 13.6% of other HCPs recalled that there were undiscussed ethical dilemmas that they thought should have been discussed. The study revealed that some HCPs rarely confront ethical dilemmas, while only a few often confront them. This issue was previously shown in our study, in which 60% of intensivists knew how to proceed if they faced ethical dilemmas and 23% had previously consulted HECs in secondary and tertiary level institutions [56]. On questions regarding the existence of HECs in their hospitals, half (51.4%) of all HCPs answered positively, meaning that they were aware of the existence of the HEC. Among physicians working in tertiary level institutions, 15 (33.3%) consulted with HECs in 2014. However, none of the physicians or other HCPs in secondary level institutions consulted with HECs in that same year. In tertiary institutions, 12 (6.9%) nurses and three (6.4%) of other HCPs had consulted with HECs in that year. However, it must be stressed that Slovenian HECs are still in the process of building up their operational capacity.

Generally, the intended role of HECs is well known (individual case consultations, education of HCPs, and policy formation) [33–35]. In line with that, the HCPs in our study, agreed that the most important roles of HECs are staff education, review of difficult cases, development of protocols, and improving communication. A very high percentage of HCPs in all groups responded that HECs are very important for resolving ethical dilemmas (84.6%) and that regular formal education about ethics is needed (87.6%). Therefore, it is not surprising that all HCPs placed regular formal education in university programmes as their first choice and learning from senior co-workers and workshop learning as their second and third choices, respectively. Core competencies and standards should be developed for healthcare ethics consultation [57].

## Limitations

Despite random sampling methods of sample representativeness, we observed underrepresentation of physicians and overrepresentation of nurses, which is related to numerically non-adequate responses in HCPs groups; this could be a source of bias in the interpretation of our results. The questionnaire was given only to HCPs, but not to the patients or laypersons accompanying or visiting the patients.

## Conclusions

In this study, we included all profiles of HCPs working in secondary and tertiary level institutions in Slovenia who are confronted with various ethical dilemmas during their daily work. The sample size was calculated accordingly to the number of all HCPs from all hospitals and with respect to the profiles of all HCPs to ensure our samples were as representative as possible. Besides pointing to well-known ethical dilemmas, our study clearly shows that waiting periods for diagnostics and treatment of the patients and suboptimal performance due to poor interpersonal relationships on the ward are ethical dilemmas that healthcare policymakers and hospital management must be made aware of. Due to their interdependence, the two ethical dilemmas probably need to be understood and solved together, at least in the Slovenian healthcare environment. The most important role of HECs is staff education, followed by improving communication, supplementing the hospital guidelines for resolving ethical dilemmas, and review of difficult cases.

## Supporting information

**S1 Appendix. Questionnaire for healthcare professionals.**
(DOCX)

**S1 File. Validation and testing of the Questionnaire.**
(DOCX)

**S1 Table. Association between the type of institution and physicians' reactions when faced with ethical dilemmas (results of univariate logistic regression with tertiary level institutions as the reference category).**
(DOCX)

**S2 Table. Association between the type of institution and nurses' reactions when faced with ethical dilemmas (results of univariate logistic regression with tertiary level institutions as the reference category).**
(DOCX)

**S3 Table. Association between the type of institution and other HCPs' reactions when faced with ethical dilemmas (results of univariate logistic regression with tertiary level institutions as the reference category).**
(DOCX)

## Acknowledgments

We thank all the Slovenian healthcare providers (physicians, nurses and all others who care for patients in the hospitals) for their participation in the study and the hospital managements for their cooperation and support.

Special thanks to Kristijan Armeni for proofreading, editing, and correcting the first draft, and to Dr Dianne Jones for proofreading the revised draft.

## Author Contributions

**Conceptualization:** Štefan Grosek, Rok Kučan, Jon Grošelj, Miha Oražem, Urh Grošelj, Jaro Lajovic, Blaž Ivanc.

**Data curation:** Štefan Grosek, Rok Kučan, Jon Grošelj, Miha Oražem, Urh Grošelj, Vanja Erčulj, Blaž Ivanc.

**Formal analysis:** Rok Kučan, Jon Grošelj, Urh Grošelj, Vanja Erčulj, Jaro Lajovic, Blaž Ivanc.

**Funding acquisition:** Štefan Grosek, Jaro Lajovic, Blaž Ivanc.

**Investigation:** Štefan Grosek, Rok Kučan, Jon Grošelj.

**Methodology:** Štefan Grosek, Rok Kučan, Jon Grošelj, Miha Oražem, Urh Grošelj, Vanja Erčulj, Blaž Ivanc.

**Project administration:** Štefan Grosek, Blaž Ivanc.

**Resources:** Štefan Grosek, Jon Grošelj, Ana Borovečki, Blaž Ivanc.

**Software:** Vanja Erčulj, Jaro Lajovic.

**Supervision:** Štefan Grosek, Rok Kučan, Jon Grošelj, Blaž Ivanc.

**Validation:** Štefan Grosek, Rok Kučan, Vanja Erčulj, Ana Borovečki, Blaž Ivanc.

**Visualization:** Rok Kučan, Vanja Erčulj.

**Writing – original draft:** Štefan Grosek, Rok Kučan, Ana Borovečki, Blaž Ivanc.

**Writing – review & editing:** Štefan Grosek, Ana Borovečki, Blaž Ivanc.

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
