## [Decision Letter · Decision Letter 0]

24 Apr 2020

PONE-D-20-08547

The first nation-wide study on facing and solving ethical dilemmas among healthcare professionals in Slovenia

PLOS ONE

Dear Dr Ivanc,

Thank you for submitting your manuscript to PLOS ONE. After careful consideration, we feel that it has merit but does not fully meet PLOS ONE’s publication criteria as it currently stands. Therefore, we invite you to submit a revised version of the manuscript that addresses the points raised during the review process.

See comments below. 

We would appreciate receiving your revised manuscript by 24 May 2020. To enhance the reproducibility of your results, we recommend that if applicable you deposit your laboratory protocols in protocols.io, where a protocol can be assigned its own identifier (DOI) such that it can be cited independently in the future. For instructions see: http://journals.plos.org/plosone/s/submission-guidelines#loc-laboratory-protocols

We look forward to receiving your revised manuscript.

Kind regards,

Andrew Soundy

Academic Editor

PLOS ONE

Journal Requirements:

Additional Editor Comments (if provided):

Thank you for this submission.

Please consider the reviewer 1 comments and respond

Please make sure you report methods according to the STROBE statement https://www.strobe-statement.org/index.php?id=available-checklists

Also please use a supplementary file for the validation of the questionnaire. So full consideration in the supplementary file and a more summarised content in the paper.

Reviewers' comments:

Reviewer's Responses to Questions

**Comments to the Author**

1. Is the manuscript technically sound, and do the data support the conclusions?

Reviewer #1: Yes

Reviewer #2: Yes

2. Has the statistical analysis been performed appropriately and rigorously? 

Reviewer #1: I Don't Know

Reviewer #2: I Don't Know

3. Have the authors made all data underlying the findings in their manuscript fully available?

Reviewer #1: Yes

Reviewer #2: Yes

4. Is the manuscript presented in an intelligible fashion and written in standard English?

Reviewer #1: No

Reviewer #2: Yes

5. Review Comments to the Author

Reviewer #1: Thank you for this study. The research topic focuses on a very crucial aspect of clinical care and physician-patient relationship. Here are my review comments:

1. Data collection was done in 2015-16 and the paper was completed and only published in 2019-20. Any reason for this large gap between data collection and paper submission? Is the study still relevant and if not what are the factors that could play role if the data collection was done between 2019-20?

2. Research question and conclusion is too ambiguous and is left to open interpretation by readers. Kindly be specific in your research question while elaborating on discussion part.

3. How were the questions in survey arrived at?

4. There is scope in the paper to offer potential solutions to the question raised. Kindly include the solution to further strengthen the paper.

Reviewer #2: This manuscript is interesting and has already been extensively reviewed. Unfortunately, in spite of its name, the responders are mostly nurses and this fact should be emphasized. I have a few other minor comments. Once these are addressed the manuscript can be accepted for publication.

6. PLOS authors have the option to publish the peer review history of their article (what does this mean?). If published, this will include your full peer review and any attached files.

Reviewer #1: Yes: Ankit Raj

Reviewer #2: Yes: Smita Neelkanth Deshpande

---

## [Author Response · Author response to Decision Letter 0]

29 May 2020

Ass. Prof. Dr. Blaž Ivanc

Faculty of Health Sciences – University of Ljubljana

Zdravstvena pot 5, SI-1000 Ljubljana,

Republic of Slovenia

24. 5. 2020

Dear Editor-in-Chief, Dr. Joerg Heber,

We wish to submit a revised version of the research article entitled “The first nation-wide study on facing and solving ethical dilemmas among healthcare professionals in Slovenia”; PONE-D-20-08547.

We attach the response to reviewers and wish to express our gratitude for carefully reviewing the manuscript.

Thank you for your consideration of the revised manuscript. 

Sincerely,

Blaž Ivanc

From: em.pone.0.6ad6c4.196d368a@editorialmanager.com <em.pone.0.6ad6c4.196d368a@editorialmanager.com> On Behalf Of PLOS ONE

Sent: Saturday, April 25, 2020 12:06 AM

To: Blaž Ivanc <ivanc.blaz@siol.net>

Subject: PLOS ONE Decision: Revision required [PONE-D-20-08547] - [EMID:d66dfb47d9c94cf7]

PONE-D-20-08547

The first nation-wide study on facing and solving ethical dilemmas among healthcare professionals in Slovenia

PLOS ONE

Dear Dr Ivanc,

Thank you for submitting your manuscript to PLOS ONE. After careful consideration, we feel that it has merit but does not fully meet PLOS ONE’s publication criteria as it currently stands. Therefore, we invite you to submit a revised version of the manuscript that addresses the points raised during the review process.

See comments below. 

We would appreciate receiving your revised manuscript by 24 May 2020. To enhance the reproducibility of your results, we recommend that if applicable you deposit your laboratory protocols in protocols.io, where a protocol can be assigned its own identifier (DOI) such that it can be cited independently in the future. For instructions see: http://journals.plos.org/plosone/s/submission-guidelines#loc-laboratory-protocols

• A rebuttal letter that responds to each point raised by the academic editor and reviewer(s). This letter should be uploaded as separate file and labeled 'Response to Reviewers'.

• A marked-up copy of your manuscript that highlights changes made to the original version. This file should be uploaded as separate file and labeled 'Revised Manuscript with Track Changes'.

• An unmarked version of your revised paper without tracked changes. This file should be uploaded as separate file and labeled 'Manuscript'.

We look forward to receiving your revised manuscript.

Kind regards,

Andrew Soundy

Academic Editor

PLOS ONE

Journal Requirements:

Reply to the Editor's comments #1:

We rechecked our manuscript to see if it meets PLOS ONE's style requirements and we can confirm that they meet these requirements, including the files names.

Reply to the Editor's comments #2:

Thank you for your comment. We added three Supporting Information Files that support the text in the manuscript:

S1_Table 1: Association between the type of institution and physicians’ reactions when faced with ethical dilemmas (results of univariate logistic regression with tertiary level institutions as the reference category)

 Secondary level institution (n = 25) Tertiary level institution (n = 51) 

 no yes no yes OR (95% CI) P-value

Discuss with head of department 8 (32) 17 (68) 11 (21.6) 40 (78.4) 0.6 (0.2; 1.7) 0.326

Discuss with colleagues 3 (12) 22 (88) 4 (7.8) 47 (92.2) 0.6 (0.1; 3) 0.559

Convene a medical council meeting 17 (68) 8 (32) 23 (45.1) 28 (54.9) 0.4 (0.1; 1.1) 0.064

Discuss with hospital medical ethics committee 24 (96) 1 (4) 33 (64.7) 18 (35.3) 0.1 (0.01; 0.6) 0.015

Discuss with national medical ethics committee (Republic of Slovenia National Medical Ethics Committee) 23 (92) 2 (8) 49 (96.1) 2 (3.9) 2.1 (0.3; 16.1) 0.463

Discuss with Legal-ethical committee of the Medical Chamber of Slovenia 25 (100) 0 (0) 51 (100) 0 (0) 

Discuss with Patient Rights Advocate 25 (100) 0 (0) 48 (94.1) 3 (5.9) 

Discuss with Human Rights Ombudsman 25 (100) 0 (0) 49 (96.1) 2 (3.9) 

Consult with hospital chaplain 24 (96) 1 (4) 51 (100) 0 (0) 

Resolve dilemma through mediation 22 (88) 3 (12) 51 (100) 0 (0) 

Contact the media 25 (100) 0 (0) 51 (100) 0 (0) 

Discuss within my family circle 19 (76) 6 (24) 48 (94.1) 3 (5.9) 5.1 (1.1; 22.3) 0.032

Decide alone 18 (72) 7 (28) 45 (88.2) 6 (11.8) 2.9 (0.9; 9.9) 0.085

* OR = odds ratio adjusted for hospital; CI = confidence interval

S2_Table 2: Association between the type of institution and nurses’ reactions when faced with ethical dilemmas (results of univariate logistic regression with tertiary level institutions as the reference category)

 Secondary level institution (n = 122) Tertiary level institution (n = 198) 

 no yes no yes OR (95% CI) P-value

Discuss with head of department 48 (39.3) 74 (60.7) 116 (58.6) 82 (41.4) 2.2 (1.4; 3.5) 0.001

Discuss with colleagues 5 (4.1) 117 (95.9) 7 (3.5) 191 (96.5) 0.9 (0.3; 2.8) 0.797

Convene a medical council meeting 119 (97.5) 3 (2.5) 194 (98) 4 (2) 1.2 (0.3; 5.6) 0.795

Discuss with hospital medical ethics committee 120 (98.4) 2 (1.6) 190 (96) 8 (4) 0.4 (0.1; 1.9) 0.246

Discuss with national medical ethics committee (Republic of Slovenia National Medical Ethics Committee) 122 (100) 0 (0) 198 (100) 0 (0) 

Discuss with Legal-ethical committee of the Medical Chamber of Slovenia 122 (100) 0 (0) 197 (99.5) 1 (0.5) 

Discuss with Patient Rights Advocate 116 (95.9) 5 (4.1) 194 (98) 4 (2) 2.1 (0.6; 7.9) 0.279

Discuss with Human Rights Ombudsman 122 (100) 0 (0) 197 (99.5) 1 (0.5) 

Consult with hospital chaplain 122 (100) 0 (0) 193 (97.5) 5 (2.5) 

Resolve dilemma through mediation 115 (94.3) 7 (5.7) 195 (98.5) 3 (1.5) 4 (1; 15.6) 0.049

Contact the media 122 (100) 0 (0) 198 (100) 0 (0) 

Discuss within my family circle 117 (95.9) 5 (4.1) 192 (97) 6 (3) 1.4 (0.4; 4.6) 0.612

Decide alone 115 (94.3) 7 (5.7) 181 (91.4) 17 (8.6) 0.6 (0.3; 1.6) 0.351

* OR = odds ratio adjusted for hospital; CI = confidence interval

S3_Table 3: Association between the type of institution and other HCPs’ reactions when faced with ethical dilemmas (results of univariate logistic regression with tertiary level institutions as the reference category)

 Secondary level institution (n = 29) Tertiary level institution 

(n = 57) 

 no yes no yes OR (95% CI) P-value

Discuss with head of department 6 (20.7) 23 (79.3) 27 (47.4) 30 (52.6) 3.4 (1.2; 9.7) 0.019

Discuss with colleagues 2 (6.9) 27 (93.1) 7 (12.3) 50 (87.7) 1.9 (0.4; 9.7) 0.447

Convene a medical council meeting 28 (96.6) 1 (3.4) 57 (100) 0 (0) 

Discuss with hospital medical ethics committee 29 (100) 0 (0) 55 (96.5) 2 (3.5) 

Discuss with national medical ethics committee (Republic of Slovenia National Medical Ethics Committee) 29 (100) 0 (0) 57 (100) 0 (0) 

Discuss with Legal-ethical committee of the Medical Chamber of Slovenia 29 (100) 0 (0) 56 (98.2) 1 (1.8) 

Discuss with Patient Rights Advocate 29 (100) 0 (0) 56 (98.2) 1 (1.8) 

Discuss with Human Rights Ombudsman 29 (100) 0 (0) 56 (98.2) 1 (1.8) 

Consult with hospital chaplain 29 (100) 0 (0) 57 (100) 0 (0) 

Resolve dilemma through mediation 29 (100) 0 (0) 57 (100) 0 (0) 

Contact the media 29 (100) 0 (0) 56 (98.2) 1 (1.8) 

Discuss within my family circle 27 (93.1) 2 (6.9) 55 (96.5) 2 (3.5) 2 (0.3; 15.3) 0.489

Decide alone 18 (62.1) 11 (37.9) 54 (94.7) 3 (5.3) 11 (2.8; 43.9) 0.001

* OR = odds ratio adjusted for hospital; CI = confidence interval

Additional Editor Comments (if provided):

Thank you for this submission.

Please consider the reviewer 1 comments and respond

Reply: Please find our reply to Reviewer # 1’s comments below.

Please make sure you report methods according to the STROBE statement https://www.strobe-statement.org/index.php?id=available-checklists

Also please use a supplementary file for the validation of the questionnaire. So full consideration in the supplementary file and a more summarised content in the paper.

Reply: Please find enclosed our supplementary files for the validation of the Questionnaire.

Reviewers' comments:

Reviewer's Responses to Questions

Comments to the Author

1. Is the manuscript technically sound, and do the data support the conclusions?

Reviewer #1: Yes

Reviewer #2: Yes

2. Has the statistical analysis been performed appropriately and rigorously? 

Reviewer #1: I Don't Know

Reviewer #2: I Don't Know

3. Have the authors made all data underlying the findings in their manuscript fully available?

Reviewer #1: Yes

Reviewer #2: Yes

4. Is the manuscript presented in an intelligible fashion and written in standard English?

Reviewer #1: No

Reviewer #2: Yes

5. Review Comments to the Author

Reviewer #1: Thank you for this study. The research topic focuses on a very crucial aspect of clinical care and physician-patient relationship. Here are my review comments:

Reply to the #1 Reviewer's comments:

We thank the reviewer for your concise review of the manuscript, which helped us to further improve the value of the manuscript.

1. Data collection was done in 2015-16 and the paper was completed and only published in 2019-20. Any reason for this large gap between data collection and paper submission? Is the study still relevant and if not what are the factors that could play role if the data collection was done between 2019-20?

Reply to the 1st comment:

The reason for this gap was that the first author (Prof. Stefan Grosek), who led the group and wrote the manuscript, was suddenly appointed Head of the Paediatric Intensive Unit when almost all of the paediatric intensivists suddenly left the PICU at the beginning of 2018 and no other experienced physician remained in the PICU. It took him almost two years to stabilise the situation and train new physicians who were able to work in the PICU. Thus, there was no time for research and academical work. We asked the Editor to extend the time for the revision of the manuscript, and he kindly agreed. In 2020, Prof. Grosek again returned to his hospital and is now able to continue his clinical, academic and research work at a slower pace, which has enabled him to complete the revision of the manuscript.

During the last few years, there have been no changes in the Slovenian health system. Therefore, these ethical dilemmas derived from our study are still valid, and very relevant. In conclusion, these results accurately and satisfactorily present common ethical dilemmas in 2015-2016, as well as today in 2020. 

2. Research question and conclusion is too ambiguous and is left to open interpretation by readers. Kindly be specific in your research question while elaborating on discussion part.

Reply to the 2nd comment: Thank you for your comment, but we cannot fully agree with you. During the first review the manuscript was already supplemented in order to clarify the research question and especially the conclusions. In addition, we have now clarified the language and hope that the text is easier to follow and understand. Please, see also our comment below and our long reply to the 2nd reviewer under item: Reply to Q20. Page 28, Line 399).

The conclusion of our reply to the 2nd reviewer was that it is not the policy of either PLOS ONE or our policy to divide our research into two or three parts and to publish them separately (we are strongly against this approach).

We tested our questionnaire in a study done at the tertiary level hospital, University Medical Centre Ljubljana. Due to the interesting results obtained in this study, we decided to extend our study to all hospitals in Slovenia. The results of this latter study are presented in this manuscript.

In conclusion, further studies are needed to evaluate in depth the answers to our questionnaires, and this will help us to draw more conclusions. Until that time, we cannot comment optimally on every result obtained, hence the evaluation of some results is left up to the readers’ interpretation.

3. How were the questions in survey arrived at?

Reply to the 3rd question: Thank you for this question. Please find our answer, which is the same as our reply to the 2nd reviewer:

The seven-step approach to questionnaire development, as recommended by the AMEE guidelines, was followed (39). We first reviewed the literature as well as including in the scope of our research the knowledge of the main ethical dilemmas found in the main tertiary hospital, University Medical Centre Ljubljana, where four of the authors are members of the Hospital Ethics Committee and daily encounter various ethical issues raised by healthcare professionals. Afterwards, we synthesised the literature and interviews and developed the questionnaire. In the next step, we included a pre-test of the questionnaire on 35 HCPs at the University Medical Centre Ljubljana (UMC Ljubljana) to optimise the measurement instrument. Based on the pre-test results, we adjusted the sample size required for measuring the primary endpoint with a predetermined precision. We also removed those questions that were not answered at all during pretesting and showed a lack of measurement sensitivity.

4. There is scope in the paper to offer potential solutions to the question raised. Kindly include the solution to further strengthen the paper.

Reply to the 4th question: Thank you for your comment. 

We strongly believe, as mentioned in the Conclusion (see lines 584-587), that the three main solutions are “staff education, followed by improving communication, supplementing the hospital guidelines for resolving ethical dilemmas, and review of difficult cases”, and they should be put in place by enhanced operations of the HECs, which have the most important role in this respective area.

To include other potential solutions to the question raised, we would need to first conduct another study in which there would be questions about possible solutions, which would then be answered by the potential participants. Without data, possible solutions presented in the manuscript would be biased towards our opinion. The health system is delicate and vulnerable, and giving some solutions in the scientific paper without having the opportunity to test them, would not be professional. However, we tried to reach our healthcare management authorities and have already internally (not-publicly) presented some our findings, at least, in the University Medical Centre Ljubljana, where some of the co-authors work.

Reviewer #2: This manuscript is interesting and has already been extensively reviewed. Unfortunately, in spite of its name, the responders are mostly nurses and this fact should be emphasized. I have a few other minor comments. Once these are addressed the manuscript can be accepted for publication.

Reply to Reviewer #2’s comments and questions

We thank the reviewer for all the comments and suggestions. As much as possible, we accepted them all, and we answered all of the questions. When we do not agree with comments, we explained in detail with our arguments why the text in question should remain unchanged in the manuscript.

Please find below all your comments and our replies.

Q1 Page 3, line 56: Full form please of HCP’s!

Reply: The full form of HCP’s included in the manuscript: Healthcare professional’s (HCP’s)

Q2. Page 4, line 76: Is this true when the research was conducted or even at time of submission of this paper?

Reply: It was true when the research was conducted, and it is still true at the time of submission of this paper. Unfortunately, nothing has changed in the intervening period.

Q3. Page 5, line 110: “online or physical should be mentioned here”

Reply: Please find the requested change in the manuscript: “We conducted a cross-sectional, physical survey-based study among HCPs (physicians, nurses…”

Q4. Page 5, line 113: “is this total or part of the total number of such institutions in the country? If only a part, how were these particular institutions chosen ( eg for existence of the committee, or large number of patients etc)?

Reply: Thank you for your question. In Slovenia, we have 14 public hospitals, three of these are tertiary University Hospitals, the University Medical Centre Ljubljana, the University Medical Centre Maribor, and Golnik University Hospital for Lung Diseases. The other 11 hospitals are secondary level general hospitals and all of these were included in our research. Other hospitals, e.g. Psychiatric hospitals, the University Rehabilitation Institute of the Republic of Slovenia and two small public maternity hospitals were not included. In conclusion, only the secondary and tertiary level large public hospitals were included.

Q5. Page 5, line114: »please describe and amplify«

Reply: Please find on Page5, Lines 127-122 a description of the selection by the proportional stratified sampling method in extenso:« Proportional stratified sampling was used to select HCPs for the study. The anonymized list of HCPs with their unique IDs was sent from each of the hospitals included in the study. We computed the proportion of HCPs to be included in the sample for each hospital. The employees included in the sample were selected based on simple random sampling (the number of seed selection units in the sample was 02031979). Simple random sampling was performed with the R software package [38] via the call to the function “sample” and with the random seed set to the date of the received list of HCPs”.

Q6. Page 5, Line 116: »have«

Reply: This was changed from “had” to “have” as suggested.

 Q7. Page 6, Line 125: »expected«

Reply: We changed “we expect” to: we expected, as suggested.

Q8. Page 6, Line 137: was a list of secondary and tertiary care hospitals made? were hospitals randomly from this list? please describe?

Reply: Please see my answer to your comment from Page 5, line 113, where we stated the number of hospitals in Slovenia (14), and how many of them are tertiary level (three hospitals) and secondary level (11 hospitals).

To be more precise, we added in the sentence: All 11 Slovenian public secondary hospitals and three tertiary level hospitals were included in the study consecutively, one by one.

Q9. Page 6, Line 141: “from the computer or otherwise?”

Reply: Yes, we chose the simple random sampling method from the computer. Please find included in the sentence: “We chose a simple random sampling method for selecting the employees who meet the criteria for inclusion in the study from the computer.

Q10. Page 6, Line 144: “delete 'follows the view that"”

Reply: This part was deleted.

Q11. Page 6, Line145: “this sentence is not clear. Names can be obtained from the employee herself, but you obtained them from each HR department.”

Reply: Thank you for this comment. The Information Commissioner stated that a public employee is not entitled to expect privacy with regard to their name. Therefore we changed the sentence in the manuscript: ” Thus, the personal information for each employee could be acquired from the Human Resources Department of each hospital, after which the employee could decide whether to participate in the survey or not.”

Q12. Page 7, Line 157: “this is more easily understood and justified”

Reply: Thank you for your comment. 

Q13. Page 7, Line 167: “delete phrase”; “delete”

Reply: All deletions were done.

Q14. Page 8, Line 192. “insert on”

Reply: I am sorry, but we do not understand what you meant by this comment. However, the Editor commented on Validation and testing the Questionnaire: “Also please use a supplementary file for the validation of the questionnaire. So full explanation in the supplementary file and a more summarised content in the paper.” 

Please, find full explanation in the supplementary file and a summarised version in the paper:

“The seven-step approach to questionnaire development, as recommended by AMEE guidelines, were followed (39). We first reviewed the literature, including in the scope of our research knowledge of the key ethical dilemmas found in the main tertiary hospital, the University Medical Centre Ljubljana, where four of the authors are members of the Hospital Ethics Committee and daily encounter various ethical issues raised by healthcare professionals. Afterwards, we synthesised the literature and interviews and developed the questionnaire. In the next step, we included a pre-test of the questionnaire on 35 HCPs at the University Medical Centre Ljubljana (UMC Ljubljana) to optimise the measurement instrument. Based on the pre-test results, we adjusted the sample size required for measuring the primary endpoint with a predetermined precision. We also removed those questions that were not answered at all during pretesting and showed a lack of measurement sensitivity. Please see the entire validation and testing of the questionnaire in the Supplementary Information file (S Validation and testing the questionnaire)”.

Q15. Page 9, Line 224, and 226: »ensured«; »was«, and Page 10, Line 230:

Reply: These corrections were made.

Q16. Page 10, Line 233:« For ease of reading, perhaps number of responses from each institution could be given as a fraction e.g. x/141: y%)”

Reply: Please find changes in the manuscript as you proposed: »The questionnaire was sent to the following 14 hospitals in Slovenia (the response rates in % and the number of questionnaires sent are in brackets for each hospital). The three tertiary level University Hospitals were: Ljubljana (52% out of 444), Maribor (44% out of 141), and Golnik (62% out of 21). The eleven secondary level general hospitals were: Topolšica (80% out of 10), Jesenice (26% out of 34), Izola (28% out of 40), Nova Gorica (41% out of 39), Novo Mesto (41% out of 48), Brežice (68% out of 19), Ptuj (80% out of 25), Murska Sobota (85% out of 40), Slovenj Gradec (38% out of 37), Celje (28% out of 86), Trbovlje (57% out of 14) The response rates in the secondary and tertiary level institutions were 45% and 51%, respectively. The final sample size was n = 485.”

Q17. Page 10, Line 235. »and how many tertiary. in your methods, please provide working definitions of these two terms for foreign readers. Whether secondary or tertiary, could be described after the name of each hospital?«

Reply: Please find working definitions of the included hospitals above in the previous reply as well as in the section Overall design of the study, Page 5).

Q18. Page 10, Line 244: »does this participation reflect the composition of the HECs, at least to some extent? What is the composition of HECs?«

Reply: If you asked, if this reflects the composition of the other HCPs, the correct answer is yes. However, if you asked if this reflects the composition of the Hospital Ethics Committee, the answer is no. We do not understand why you asked if this composition reflects the composition of the HEC's? The composition of HEC's was not one of our research questions. However, we think it is an important issue, and it is a matter of further research (our research group developed a special questionnaire that only deals with the functioning of HEC's, but this is a subject of a planned special research, which in on-going…). 

Q19. Table 1:« perhaps, the majority group in each category could be highlighted/”

Reply: We followed your suggestion.

Q20. Page 28, Line 399 Discussion:«remark: the results section is overly long due to which the importance of several findings tends to get lost. Since the extensive tables provide the minuter details, can the text focus more on significant findings and point to relevant table for further details?«

Reply: Thank you very much for raising this issue. We are not convinced that the Results section is overly long. This study was very comprehensive and touches many important questions and answers that physicians, nurses, and other HCPs face during their daily work in the hospital. The questionnaire form is, in our opinion, well structured and it allows the participants to respond to vastly different situations and problems they face in the hospital setting. It was validated and first tested among HCPs at the largest hospital in Slovenia, the University Medical Centre Ljubljana. Only later did we continue our study in other Slovenian hospitals.

We also carefully read our Discussion again and we also asked the authors who participated in this study to comment on this after re-reading the manuscript. Most of them are very knowledgeable and experienced and highly educated persons who finally approved this manuscript in the form in which it was sent to the PlosOne journal. We also asked for the manuscript to be proofread by a native speaker and translator, Dr Dianne Jones.

Some of the researchers would prefer to divide this study into two or three parts and present this part separately in two or three manuscripts. We, as authors, are strongly against the division of the manuscript into several parts.

Finally, we reread the Discussion section, with the recent changes of the text, and found that the results in the Result section are explained concisely, so that even those who are not familiar with the subject matter can easily understand the results. We believe that we were able to show differences between three different groups of HCPs, we showed how different groups of HCPs solved ethical dilemmas, and, finally, this study enabled us to make the health authorities aware of the main ethical problems in Slovenian hospitals.

Q21. Page 28, Line 410:« do you mean 'whereas'?«

Reply: This was corrected.

Q22. Page 29, Line 430: »however please comment on rate of response to the survey which varied from 40% to higher. Also please write your discussion in paragraphs rather than one continuous paragraph.«

Reply: This question has already been raised by the former reviewer, and we answered that we do not know the cause of the high variability in the response rates.

Concerning your second comment, we will follow your suggestion.

Q23. Page 29, Line439: »what about nurses- the vast majority in your survey? It is interesting to note that they seemed to experience fewer dilemmas and fewer still consulted the ethics committee. This is unusual because the nursing curriculum does include ethics.»

Reply: Thank you for your comment. That was indeed a surprise to us, and it will require further research among nurses. We hope that we will be able to conduct further research on this matter.

Q24. Page 29, Line 443:« as i remarked before, did the composition or working style of the HEC have something to do with this result?«

Reply: This indeed could have something to do with these results. Due to these results, we want to conduct another study among members of HECs in Slovenia to see how they work and to identify the main ethical issues that they have to face and deal with.

Q25: Page 30. Lines 475, 478, 490, and Page 32, Line 520: ”delete”, and “repetition”.

Reply: These corrections were made in the revised manuscript.

Q27. Page 32, Line524: “discussing medical dilemmas with family members is itself an ethical issue around confidentiality!”

Reply: We completely agree with the Reviewer’s comment. However, it is also true that we discuss ethical issues we are involved in with our closest family members. While preparing the questionnaire, the research group asked themselves whether to include this question or not. Unanimously, the answer was yes. We believe that this phaenomena should be subjected to further research, which is very demanding and calls for multi-disciplinary research approach.

Q28. P33, Line 561, “half”

Reply: This was corrected.

---

## [Decision Letter · Decision Letter 1]

17 Jun 2020

The first nation-wide study on facing and solving ethical dilemmas among healthcare professionals in Slovenia

PONE-D-20-08547R1

Dear Dr. Ivanc,

We’re pleased to inform you that your manuscript has been judged scientifically suitable for publication and will be formally accepted for publication once it meets all outstanding technical requirements.

Kind regards,

Andrew Soundy

Academic Editor

PLOS ONE

Additional Editor Comments (optional):

Reviewers' comments:

Reviewer's Responses to Questions

**Comments to the Author**

1. If the authors have adequately addressed your comments raised in a previous round of review and you feel that this manuscript is now acceptable for publication, you may indicate that here to bypass the “Comments to the Author” section, enter your conflict of interest statement in the “Confidential to Editor” section, and submit your "Accept" recommendation.

Reviewer #1: All comments have been addressed

Reviewer #2: All comments have been addressed

2. Is the manuscript technically sound, and do the data support the conclusions?

Reviewer #1: Yes

Reviewer #2: Yes

3. Has the statistical analysis been performed appropriately and rigorously? 

Reviewer #1: Yes

Reviewer #2: I Don't Know

4. Have the authors made all data underlying the findings in their manuscript fully available?

Reviewer #1: Yes

Reviewer #2: Yes

5. Is the manuscript presented in an intelligible fashion and written in standard English?

Reviewer #1: No

Reviewer #2: Yes

6. Review Comments to the Author

Reviewer #1: (No Response)

Reviewer #2: Your firm yet polite refusal to accept all reviewer comments, with proper explanations, is appreciated!

7. PLOS authors have the option to publish the peer review history of their article (what does this mean?). If published, this will include your full peer review and any attached files.

Reviewer #1: Yes: Ankit Raj

Reviewer #2: No

---

## [Editor Report · Acceptance letter]

23 Jun 2020

PONE-D-20-08547R1 

The first nationwide study on facing and solving ethical dilemmas among healthcare professionals in Slovenia 

Dear Dr. Ivanc:

I'm pleased to inform you that your manuscript has been deemed suitable for publication in PLOS ONE. Congratulations! Your manuscript is now with our production department. 

Kind regards, 

on behalf of

Dr. Andrew Soundy 

Academic Editor

PLOS ONE